# Negative Pressure Wound Therapy with Instillation: Analysis of the Rinsing Fluid as a Monitoring Tool and Approach to the Inflammatory Process: A Pilot Study

**DOI:** 10.3390/jcm12020711

**Published:** 2023-01-16

**Authors:** Niklas Biermann, Stefan Wallner, Teresa Martini, Steffen Spoerl, Lukas Prantl, Christian D. Taeger

**Affiliations:** 1Department of Plastic, Hand and Reconstructive Surgery, University Hospital Regensburg, 93053 Regensburg, Germany; 2Department of Clinical Chemistry, University Hospital Regensburg, 93053 Regensburg, Germany; 3Clinic and Polyclinic for Oral and Maxillofacial Surgery, University Hospital Regensburg, 93053 Regensburg, Germany

**Keywords:** negative pressure wound therapy with instillation, inflammation, wound healing

## Abstract

Background: Negative pressure wound therapy with instillation (NPWTi) is an established wound conditioning tool. Previous investigations discovered that the rinsing fluid is a suitable monitoring tool containing various cells and cytokines. Methods: The aim of this pilot study was to analyze rinsing fluid samples from patients treated with NPWTi and link them to the clinical course, including microbiological contamination. In 31 consecutive patients with acute and chronic wounds, laboratory analysis was performed to evaluate IL-6, IL-8, bFGF, Tnf-a, and VEGF. Results: IL-6 showed a significant increase to 1540 pg/mL on day two and 860 pg/mL on day four (*p* = 0.01 and *p* = 0.04, resp.). IL-8 steadily increased from a median of 2370 pg/mL to a maximum of 19,400 pg/mL on day three (*p* = 0.01). The median bFGF showed a steady decline from 22 pg/mL to 10 pg/m (*p* = 0.35) on day three. The median Tnf-a increased from 11 pg/mL to 44 pg/mL (*p* = 001). The median VEGF values fluctuated but showed an overall increase from 35 pg/mL to 250 pg/mL (*p* = 0.07). Regarding IL-8, diabetic and non-diabetic patients both showed a gradual increase with non-significant higher median values for the diabetics. The subgroup analysis of IL-6 showed increasing and higher values in cases with bacterial superinfections (*p* = 0.07). Conclusion: We were able to use an established wound conditioning tool to gather important information about the inflammatory response during NPWTi treatment. Cytokine and cell courses were mostly consistent with the literature, especially in diabetic patients, and should be further investigated.

## 1. Introduction

Wound healing is a highly orchestrated and overlapping interaction of cells and cytokines involving inflammation, tissue production, and remodeling. Impaired cutaneous wound healing is a common cause for inpatient treatment and, in some cases, surgical debridement and reconstruction. In patients with comorbidities and impaired immune defenses, superinfections or delayed skin closure requires wound bed conditioning with negative pressure wound therapy and additional instillation or other topical treatments [1,2,3,4,5,6,7]. Literature suggests that instillation is a key factor in achieving superior results by reducing both the bioburden and accelerating wound healing. The underlying mechanism of action is still poorly understood, as well as the influence on the inflammatory cascade [8,9,10,11,12]. Additionally, wound evaluation during the dressing phase is impossible, and monitoring tools are lacking. With the increasing prevalence of comorbidities such as peripheral artery disease or diabetes, understanding the pathophysiology of the disease and its treatment modality are crucial [13,14].

Recent studies have shown that wound fluid contains locally produced and absorbed products, which provide insight into the wound-healing cascade [15,16,17]. Conventional negative pressure therapy in open abdominal wounds showed distinct clearance patterns of cytokines when compared with serum levels, thus allowing the investigation of abdominal sepsis [18]. Other studies evaluated the rinsing fluid of NPWT(i) devices and found that basic laboratory markers such as pH, electrolytes, and flow cytometry could safely be determined [16]. Increasing levels of polymorph nuclear cells marked the beginning of the inflammatory healing phase, and increased potassium values resembled cell destruction after surgical debridement [16]. However, benchmark values for the rinsing fluid are lacking, and further investigation seems promising.

Therefore, we analyzed previously gathered frozen fluid samples from NPWTi devices over several days for patients with acute and chronic wounds. The aim of this pilot study was to establish cytokine profiles that allow for evaluation of the wound during the no-view treatment phase and correlate these with patient characteristics and bacterial bioburden. The physician might then be able to react more accordingly, for example, change a dressing if rising cytokines indicate an infection. Up to now, the cytokine content within the rinsing fluid has not been investigated, so we chose IL-6, IL-8, bFGF, Tnf-a, and VEGF since these depict the central agents with respect to chemotaxis, inflammation, and fibroblast recruitment within cutaneous wound healing.

## 2. Materials and Methods

This is a prospective observational study investigating different aspects of wound healing during NPWTi by analyzing the rinsing fluid. It was conducted with ethics approval and according to the 2010 CONSORT (CONsolidated Standards of Reporting Trials) guideline based on the declaration of Helsinki. It was registered in a public German trial registry (DRKS00017669).

We report on an existing patient collective investigated between September 2019 and May 2020 [16]. A total of 31 consecutive patients with open wounds needing surgical debridement and NPWTi were included regardless of the cause of the wound, except for cancer. Samples were taken each day during the inpatient treatment. For each surgical intervention, a swab was also sent for microbiological assessment. These specimens are further analyzed in the current study.

Initially, fluid samples were taken from a special sampling canister from the NPWT(i) device (V.A.C. VERAFLO^TM^, KCI Medizinprodukte GmbH, Wiesbaden, Germany) using a sterile syringe and immediately sent to the laboratory. The instillation fluid was a standard 0.9% sodium chloride solution (Braun, Melsungen, Germany).

Five Eppendorf cups containing 0.5 mL of sample fluid were frozen at −80 °C for further analysis in commercially available CLIAs (Immulite1000, Siemens Healthineers, Erlangen, Germany). These samples were used for the current quantitative measurements of IL-6 and IL-8 using electrochemiluminescence immunoassay (cobas e411, Roche Diagnostics, Basel, Switzerland) and chemiluminescence immunoassay (Dimension Vista 1500, Siemens Healthineers, Erlangen, Germany), respectively. Additionally, we also analyzed Tnf-a, VEGF, and bFGF (Immulite1000, Siemens Healthineers, Erlangen, Germany). All measurements were performed according to the manufacturer’s instructions. To ensure accuracy, the analysis was performed in an accredited university laboratory during the daily routine by specialists of the Department of Clinical Chemistry. Prior to each measurement, a validation dilution graph was created for each cytokine using the manufacturer’s sample to ensure the correct setup of the luminescence systems. With each measurement, a control NaCl sample was used for validation, and multiple tests were repeated. Statistical analysis was performed using the Wilcoxon test for nonparametric paired values over several days, the Mann–Whitney U test for nonparametric values on the same day, and the Kruskal–Wallis test for comparison of more than two medians.

A *p*-value <0.05 indicated significance, and the “Statistical Package for the Social Sciences” (SPSS Inc., Chicago, IL, USA) Version 25.0 was utilized for calculation.

## 3. Results

A total of 31 patients with an overall mean age of 63 (SD 16.9) were included in the study. A total of 20 (64.5%) were male, and 11 (35.5%) were female. The reason for wound treatment was chronic in 23 patients (ischemic *n* = 2, post-traumatic/post-infection *n* = 12, decubital ulceration *n* = 6, post-cancer *n* = 3) and acute in eight patients (traumatic *n* = 4, burns *n* = 1, ischemic/compartment *n* = 3). Of the chronic cases, diabetes was etiological in six and peripheral artery disease (PAD) in eight patients. The microbiological workup revealed concomitant bacterial or fungal infections in 24 cases, and 7 patients showed sterile microbiological results Table 1**.**

The standardized laboratory fluid analysis included measurements of IL-6, IL-8, bFGF, Tnf-a, and VEGF. A preliminary analysis was performed, then the overall values of each parameter were analyzed over the first four days after the initial operation.

IL-6 showed a fluctuating course with a significant median increase to 1540 pg/mL on day two and 860 pg/mL on day three (*p* = 0.01 and *p* = 0.04, respectively) compared to the first measurement Figure 1A.

IL-8 steadily increased from a median of 2370 pg/mL on the operating day to a maximum of 19,400 pg/mL on day three *p* = 0.01 Figure 1B.

The median bFGF showed a steady, non-significant decline from 22 pg/mL on the operating day to 10 pg/mL (*p* = 0.35) on day three Figure 1C.

In contrast, the median Tnf-a value steadily increased from 11 pg/mL in the first measurement to 44 pg/mL on day three (*p* = 0.01) Figure 1D.

The median VEGF values fluctuated on days one and two but showed an overall increase from 35 pg/mL on the operating day to 250 pg/mL on day three *p* = 0.07 Figure 1E.

To further differentiate between the various courses and investigate whether the values might be linked to the clinical situation, the following groups were created: diabetics, PAD, and microbiological workup.

With respect to IL-8, diabetic and non-diabetic patients both showed a gradual increase with higher median values than the diabetics in the first three measurements *p* = 0.73, *p* = 0.45, and *p* = 0.40, respectively Figure 2.

In the group of patients with bacterial and fungal infections, IL-8 showed a steady increase in the presence of bacteria with higher median values of 11,980 pg/mL and 27,000 pg/mL on days two and three compared to 5220 pg/mL and 1180 pg/mL in patients with no superinfection, *p* = 0.93 and *p* = 0.05, respectively Figure 3.

Simultaneously, polymorphnuclear cells increased in patients with a bacterial infection, but no significant difference was noted compared to no infection and fungal contamination (*p* = 0.13) Figure 4.

The subgroup analysis of IL-6 showed increasing and higher values in cases of bacterial superinfections on days two and three, *p* = 0.94 and *p* = 0.07, respectively Figure 5. 

Diabetic patients showed no significant differences in the first three measurements (*p* = 0.63, *p* = 1.0, and *p* = 0.76, respectively) but lower values on day three with 206 pg/mL compared to 1730 pg/mL, *p* = 0.26 in non-diabetic patients Figure 6.

Regarding VEGF, patients with PAD had higher levels on the day of the first surgery, with 100 pg/mL compared to 32 pg/mL in non-PAD patients, *p* = 0.45. Over time there was a gradual decline with lower levels on days two and three, *p* = 0.04 and *p* = 0.12, respectively Figure 7.

A similar course was noted for diabetic patients with initially elevated levels of 100 pg/mL compared to 32 pg/mL in non-diabetic patients, *p* = 0.84. A decline with lower levels was noted on days two and three, *p* = 0.88 and *p* = 0.34, respectively Figure 8.

The subanalysis of bFGF in diabetic and non-diabetic patients showed higher median values in diabetic patients in every measurement starting from the operating day until day three, *p* = 0.66, *p* = 0.54, *p* = 0.71, and *p* = 1.0, respectively Figure 9.

A similar course was noted for Tnf-a levels in diabetic patients who showed higher Tnf-a levels compared to non-diabetics in every measurement until day three, *p* = 0.54, *p* = 0.12, *p* = 0.54, and *p* = 0.56, respectively Figure 10.

With respect to Tnf-a and patients with bacterial superinfections, there was a steady increase with higher values compared to no superinfection on days two and three, *p* = 0.56 and *p* = 0.06, respectively Figure 11.

## 4. Discussion

Wound healing follows highly orchestrated inflammatory pathways involving various cytokines, cells, and immunological processes. NPWTi is a modern wound dressing technique that is successfully used in daily clinical practice for the treatment of acute and chronic wounds. With this study, we wanted to test whether it is possible to gain insight into the wound-healing process from the extracted irrigation fluid. In addition, we wanted to test whether different patient cohorts show different profiles in the analysis.

We found consensus in the literature for the concentrations of our outcome parameters, proving the accuracy of our methodology as a monitoring tool during the dressing phase. One of the main chemotactic agents for polymorphnuclear cells throughout the inflammatory wound-healing phase is IL-8. Secreted by endothelial cells, keratinocytes, and fibroblasts, it has a direct influence on keratinocyte migration and adhesion, thereby significantly influencing wound healing. In chronic wounds, subunits of the IL-8 receptor, IL-8RB, are markedly decreased and thus limit its function [19], plus direct wound treatment with topical IL-8 in mice showed enhanced reepithelization [20].

In diabetic ulcers, Meng et al. investigated the wound exudate and found significantly increased levels of IL-8 up to 600 pg/mL. Additionally, in vitro stimulation of forearm skin fibroblasts with advanced glycation end products led to a 30-fold increase in IL-8 production when compared to their control [21].

We can confirm these findings since patients in our cohort with diabetes tended toward higher IL-8 levels compared to non-diabetic patients Figure 2. With respect to the chemotactic properties, there was a steady increase in the IL-8 course and non-significant, higher values on days two and three for wounds with bacterial superinfections. Interestingly, fungal infections showed even lower IL-8 values when compared to no infection at all Figure 3.

Particularly in the context of antigen presence, another important central pro-inflammatory cytokine is IL-6. It is produced by keratinocytes as well as both fibroblasts and mononuclear cells as a response to a certain antigen interaction. After connection with the membrane-bound IL-6 receptor, lymphocytes are stimulated, and acute-phase proteins are produced. This sequence can be confirmed by our results since bacterial superinfections led to higher IL-6 levels compared to non-infected wounds Figure 5. Again, fungal infections showed equal or even lower IL-6 levels.

With regard to cutaneous wound healing, IL-6 plays a pivotal role [22,23], similar to pre-existing diabetes. Lee et al. investigated IL-6 production in the presence of hyperglycemia. They found delayed wound healing in diabetic mice with the higher secretion of IL-6 protein and its receptor in the wound when compared to their control [24]. In our cohort, patients with diabetes showed no significant differences in the secreted IL-6 levels Figure 6. However, the measured values were very similar to Lee et al. ranging up to about 1200 pg/mg [24].

Especially during the remodeling of wounds and in patients with PAD, VEGF-A is a key regulator of angiogenesis, which undergoes splicing into different isoforms. In PAD, plasma levels of VEGF are elevated, but it has been shown that anti-angiogenic isoforms of VEGF-A are over-expressed, and pro-angiogenic forms are under-expressed [25,26]. We can confirm that PAD and diabetic patients had higher VEGF-A levels on the day of the first surgery but noted a gradual decline to lower levels on days two and three. Possibly the debridement of poorly vascularized and partly necrotic tissue might lower the angiogenic stimulus, or the generally less vascular endothelial cells are not at an accountable level during the inflammatory wound phase.

Once the inflammatory phase is over and the extracellular matrix is built, bFGF is a prominent member of the growth factor family responsible for cell migration, growth, and differentiation. In diabetes, however, it has been shown that delayed wound healing and especially low cell migration can be linked to the inhibition of the bFGF signaling pathway [27]. Thus, higher compensatory levels seem logical and can be confirmed by our results in diabetic patients Figure 9.

Using negative pressure wound therapy, Yang et al. showed a significant up-regulation of the bFGF gene in diabetic patients when compared to conventional gauze therapy [4]. As in our study, all patients were treated with NPWT, but changes were also noted in patients not suffering from diabetes; thus, we cannot confirm these findings.

Another central cytokine predominantly produced by macrophages and an inducer of a pro-inflammatory reaction is Tnf-a. In wound repair, it plays a variety of roles in cell interaction. Sequeira et al. investigated the role of Tnf-a with respect to fibroblasts and wound healing in diabetic mice. They found increased rates of fibroblast apoptosis and Tnf-a levels of 300 pg/mL compared to 100 pg/mL in diabetic and non-diabetic mice, respectively. A systemic Tnf-a inhibition led to less apoptosis and increased wound healing [28]. We can confirm these findings as well since patients with diabetes showed higher Tnf-a levels in every measurement Figure 10.

Fahey et al. investigated the ability of fibroblasts to produce Tnf-a in response to LPS. They found that only fibroblasts with this ability were harvested from wounds, as compared to normal subcutaneous fibroblasts, suggesting that fibroblasts have unique features with different phenotypes [29]. In our investigation, there was a steady Tnf-a increase with higher values on days two and three with bacterial superinfections Figure 11. Again, the inflammatory reaction with a concomitant fungal infection was different, as seen in lower Tnf-a levels.

This study has several strengths and limitations. The primary strength is the prospective design and use of pre-existing and well-established measuring tools. The fluid samples were immediately processed and frozen for this analysis. Scaling problems have previously been ruled out by significant daily changes in the total protein amount and the proportional fluid application of the NPWT(i) device. However, a decreasing and heterogeneous number of fluid samples following day three limited the statistical power in the long-term analysis. In general, although the establishment of benchmark values was not possible, we were able to confirm the concentrations of several outcome parameters in the literature.

In conclusion, we were able to use an established wound-treating tool to gather important information about the wound-healing status during the dressing phase. Cytokine and cell courses were mostly consistent with the literature, especially in diabetes, and therefore should be investigated in combination with artificial intelligence in further work. Physicians should be able to determine the timing of the dressing change and detect superinfections by analyzing the rinsing fluid. Similar technologies, such as conventional NPWT, could be investigated further as well since wound fluid is also available with this methodology. Finally, larger cohorts should be investigated for long-term profiles and further differentiation of bacterial and fungal infections.

## Figures and Tables

**Figure 1 jcm-12-00711-f001:**
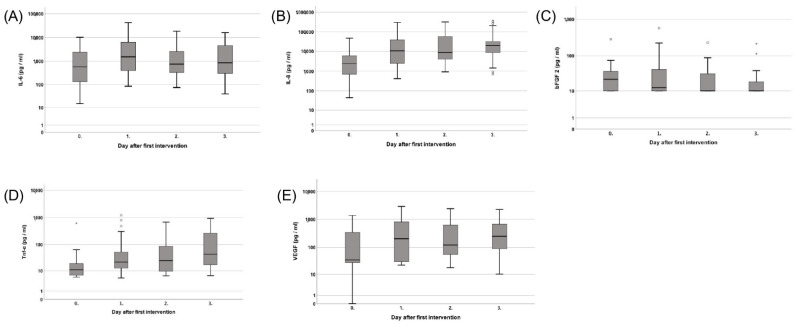
(**A**) Course of Interleukin 6 (IL-6), (**B**) Interleukin 8 (IL-8), (**C**) Basic Fibroblast Growth Factor 2 (bFGF 2), (**D**) Tumor Necrosis Factor alpha (Tnf-a), and (**E**) Vascular Endothelial Growth Factor (VEGF) over the first four measurements. Asterisk (*) indicates significance.

**Figure 2 jcm-12-00711-f002:**
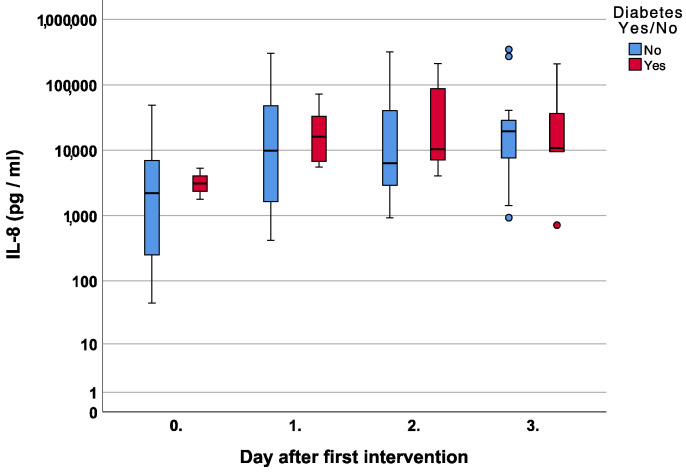
Course of Interleukin 8 (IL-8) over the first four measurements in patients with and without diabetes.

**Figure 3 jcm-12-00711-f003:**
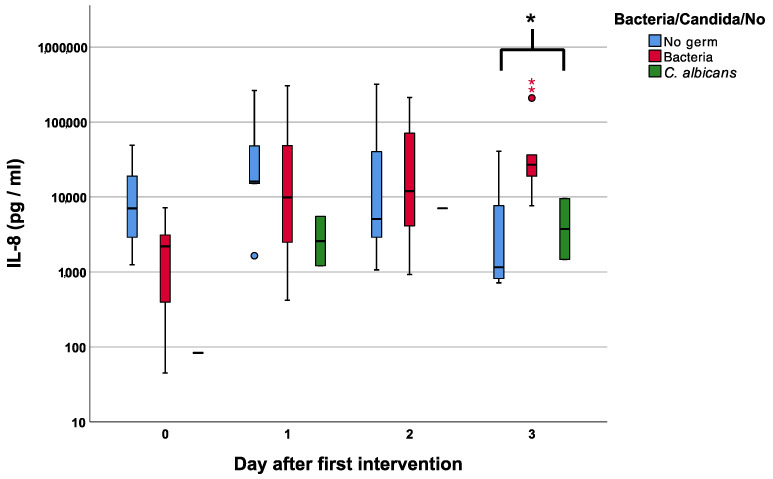
Course of Interleukin 8 (IL-8) over the first four measurements in patients with bacterial, fungal, and no wound contamination. Asterisk (*) indicates significance.

**Figure 4 jcm-12-00711-f004:**
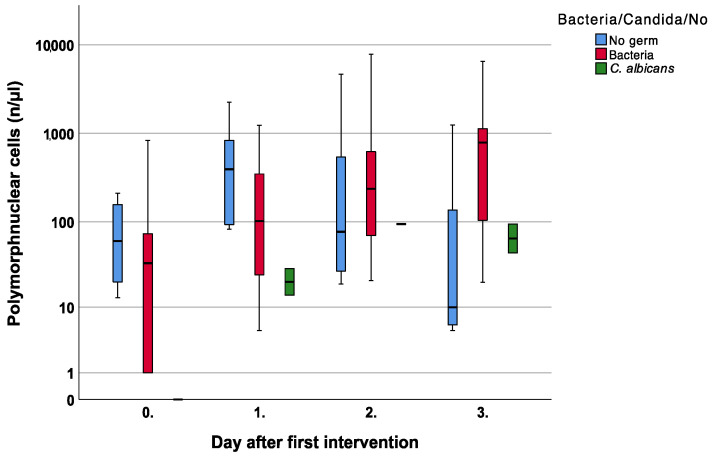
Course of polymorphnuclear cells over the first four measurements.

**Figure 5 jcm-12-00711-f005:**
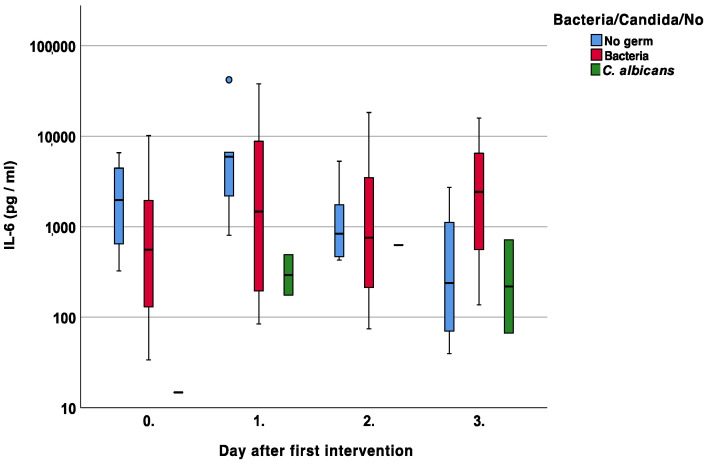
Course of Interleukin 6 (IL-6) over the first four measurements in patients with bacterial, fungal, and no wound contamination.

**Figure 6 jcm-12-00711-f006:**
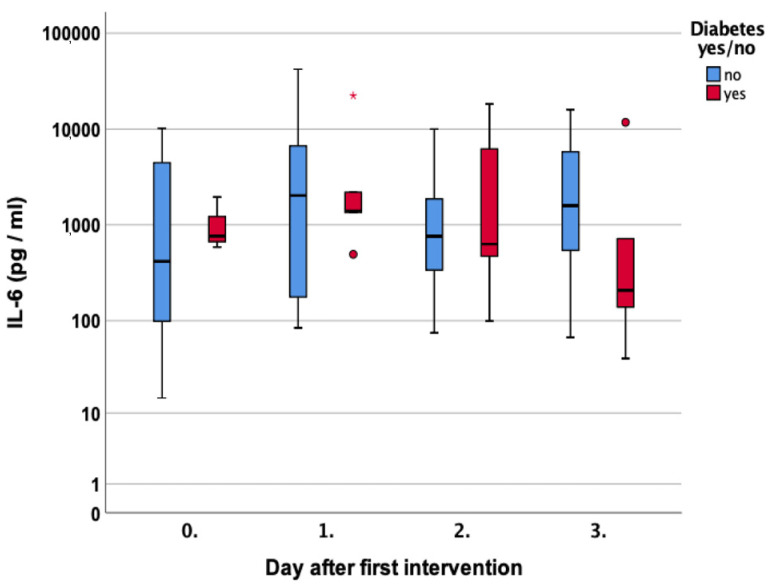
Course of Interleukin 6 (IL-6) over the first four measurements in patients with and without diabetes. Asterisk (*) indicates significance.

**Figure 7 jcm-12-00711-f007:**
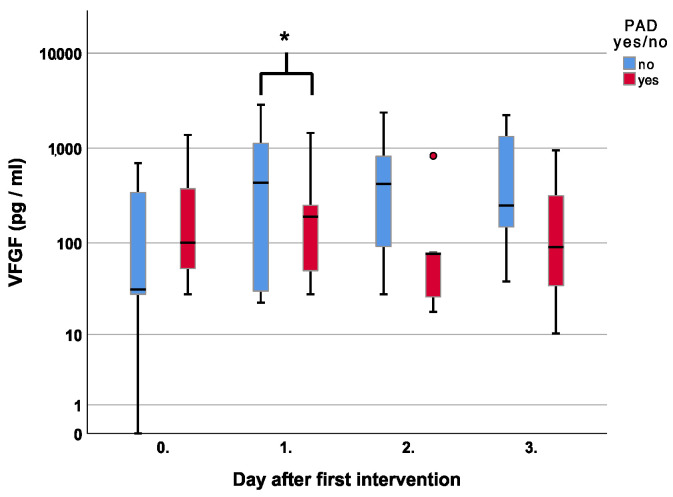
Course of Vascular Endothelial Growth Factor (VEGF) over the first four measurements in patients with and without peripheral artery disease. Asterisk (*) indicates significance.

**Figure 8 jcm-12-00711-f008:**
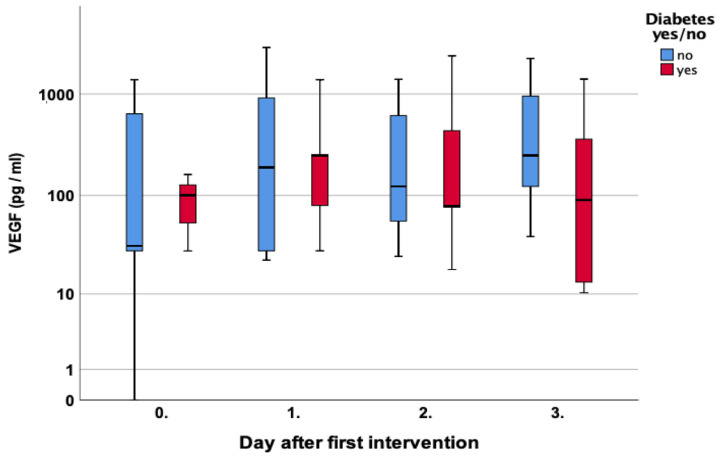
Course of Vascular Endothelial Growth Factor (VEGF) over the first four measurements in patients with and without diabetes.

**Figure 9 jcm-12-00711-f009:**
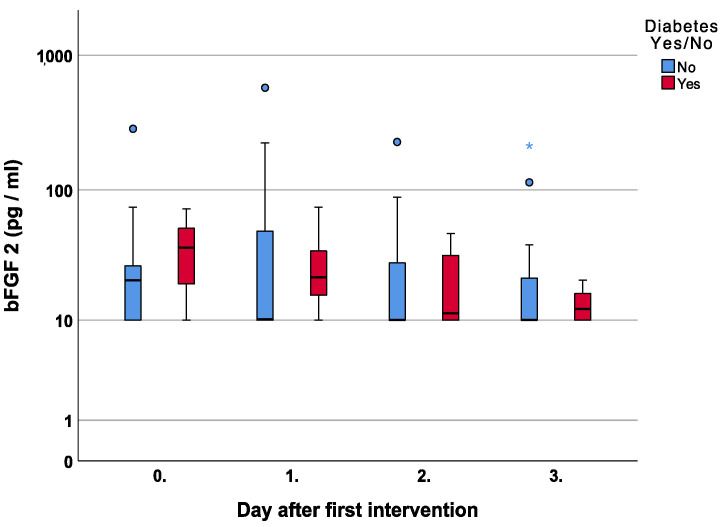
Course of Basic Fibroblast Growth Factor 2 (bFGF 2) over the first four measurements in patients with and without diabetes. Asterisk (*) indicates significance.

**Figure 10 jcm-12-00711-f010:**
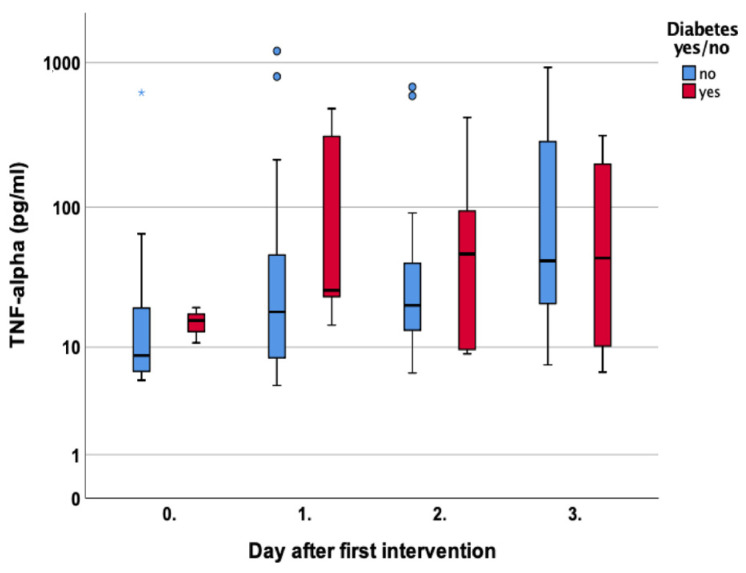
Course of Tumor Necrosis Factor alpha (Tnf-a) over the first four measurements in patients with and without diabetes. Asterisk (*) indicates significance.

**Figure 11 jcm-12-00711-f011:**
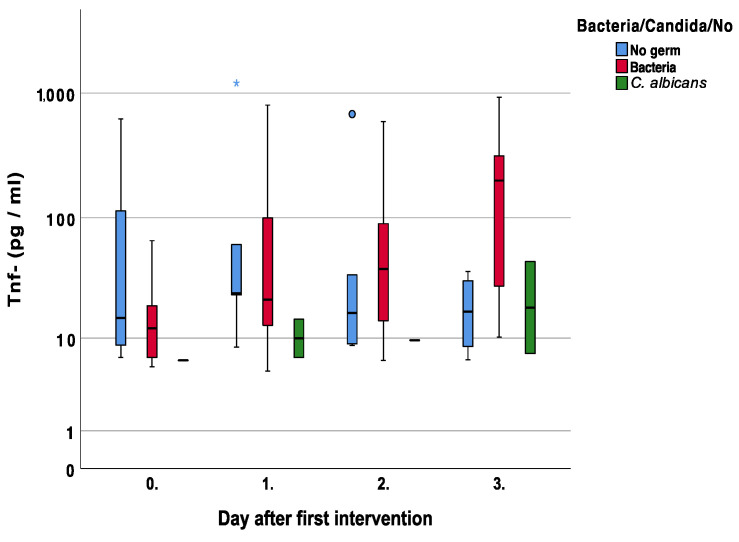
Course of Tumor Necrosis Factor alpha (Tnf-a) over the first four measurements in patients with bacterial, fungal, and no wound contamination. Asterisk (*) indicates significance.

**Table 1 jcm-12-00711-t001:** Patient characteristics.

**Patients**			
**Sex**	**Male *n* = 20**	**Female *n* = 11**	
**Reason**	**Chronic *n* = 23** Ischemic *n* = 2 Post-traumatic/post-infection *n* = 12 Decubital ulceration *n* = 6 Carcinoma-associated *n* = 3 **Diabetes = 6** **Peripheral Artery Disease = 8**	**Acute *n*= 8** Traumatic *n* = 4 Ischemic/compartment *n* = 3 Burns *n* = 1	
**Microbiological** **contamination**	**Bacteria** *S. aureus n* = 10 *MRSA n* = 2 *S. epidermidis n* = 4 *S.pyogenes n* = 2 *S*. *anginosus/massilensis n* = 1 *E. coli n* = 3 *E. faecalis n* = 6 *P. aeruginosa n* = 3 *Proteus spp*. *n* = 4 *Klepsiella spp.* *n* = 2 Polymicrobial (≤2) *n* = 5 Polymicrobial (≥3) *n* = 2	**Fungus** *Candida albicans n* = 3	**No contamination** *n* = 7

## Data Availability

Data is contained within the article.

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
