# Peer review of "Negative Pressure Wound Therapy with Instillation: Analysis of the Rinsing Fluid as a Monitoring Tool and Approach to the Inflammatory Process: A Pilot Study"

_jcm, 2023, doi:10.3390/jcm12020711_

Round 1
Reviewer 1 Report
The manuscript investigated “Negative pressure wound therapy with instillation: analysis of the rinsing fluid as a monitoring tool and approach to the inflammatory process. A pilot study.” It is a meaningful work, but I think the manuscript still has alittle problems and needs to do more work to improve the quality of this manuscript. Under the present state, I therefore suggest a Minor revision of the manuscript. I have added a detailed list of comments below: they should be taken into account by the authors when reworking the manuscript.
1. Lines 57 and 58 have problems..
2. The following reference could support the Introduction section:
Mousavi, Seyyed-Mojtaba, Zohre Mousavi Nejad, Seyyed Alireza Hashemi, Marjan Salari, Ahmad Gholami, Seeram Ramakrishna, Wei-Hung Chiang, and Chin Wei Lai. "Bioactive agent-loaded electrospun nanofiber membranes for accelerating healing process: A review." Membranes 11, no. 9 (2021): 702.
3. The English language must be carefully revised by a Native english Speaker.
4. Please, reference format corrected.
5. How was the accuracy of the analysis evaluated?
6. Under what conditions was the analysis done?
Author Response
Dear Editor, dear Reviewer,
Again, we would like to thank the reviewers for the helpful comments. We meticulously worked through the review and hope to meet the demands of the reviewers. We added the proposed comments and a few references to more explain our introduction, methodology and conclusion.
In the following paragraphs, we would like to take the opportunity to address all comments by the reviewers in a point-by-point manner (the changes we made to the manuscript are highlighted in red letters in the revised version):
Point-by-point reply to the reviewer’s comments:
Reviewer 1:
1.Lines 57 and 58 have problems..
Author reply: We apologize for the imprecise wording and corrected the sentence accordingly.
- The following reference could support the Introduction section:
Mousavi, Seyyed-Mojtaba, Zohre Mousavi Nejad, Seyyed Alireza Hashemi, Marjan Salari, Ahmad Gholami, Seeram Ramakrishna, Wei-Hung Chiang, and Chin Wei Lai. "Bioactive agent-loaded electrospun nanofiber membranes for accelerating healing process: A review." Membranes 11, no. 9 (2021): 702.
Author reply: Thank you very much for this suggestion, we read the mentioned article and added it to the manuscript.
- The English language must be carefully revised by a Native english Speaker.
Author reply: Thank you very much for this comment, the whole article was read by a native speaker and carefully revised.
Please, reference format corrected.
Author reply: Thank you for this suggestion. The reference format has now been determined by the journal and unified in the revised version. We hope to meet your expectations.
How was the accuracy of the analysis evaluated?
Author reply: Thank you very much for this notation. We apologize for the brief Methods section and added it accordingly. To ensure accuracy, the analysis was done in an accredited university laboratory during the daily routine by specialist of the department of Clinical Chemistry. Prior each measurement, a validation dilution graph was created for each cytokine using the manufacturer’s sample to ensure the correct setup of the luminescence systems. With each measurement, a control NaCl sample was tested for validation and multiple tests were repeated. A p-value <0.05 was chosen to indicate significance and the “Statistical Package for the Social Sciences” (SPSS Inc., Chicago, IL, USA) was utilized for calculation.
Under what conditions was the analysis done?.
Author reply: Thank you for this comment. We tried to outline the conditions and methods more under question 5 and added the manuscript accordingly.
To ensure accuracy the analysis was done in an accredited university laboratory during the daily routine by specialist of the department of Clinical Chemistry.
Reviewer 2 Report
In this manuscript, Biermann et al. have measured inflammatory cytokine and reparative growth factor levels in rinsing fluid from patients that underwent NPWTi treatment to demonstrate its utility as a monitoring tool, as well as to establish baseline trends in these molecular markers with the progression of treatment. This works continues from the work published previously that established the metabolite and cellular composition of NPWTi rinsate. Overall, the study is well presented, and demonstrates rigor that advances the field forward.
Major remarks:
1. The introduction is bare-bones, and needs to be expanded more to help orient the reader better on the topic. Suggested items to include are:
a. Why were the cytokines that were studied in this manuscript chosen (roles in inflammation, repair, etc.)?
b. How does the surgeon/physician benefit from monitoring the rinsing fluid cytokine levels?
c. Include and comment on other studies such as Faes et al. (PMCID: PMC7790174) that has measured inflammatory cytokine levels or cellular composition during conventional negative pressure therapy, also highlighting how this study differs from them.
d. How do co-morbidities and microbial infection play into NPWT therapy and wound healing?
2. It is suggested that the number of figures be brought down from 15, by clubbing several figures together as panels of one figure. For instance, Figs. 1-5 can be made into 5 panels of Fig. 1.
3. In the discussion, can you hypothesize/comment on why rinsate from non-germ wounds show higher Day 0 IL-8, IL-6 and TNF-a, compared to bacterial infected wounds? This is against the expectation that infected wounds would attract greater immune response and inflammation.
4. This study establishes the usefulness of rinsing fluid as a marker for healing progression. Therefore, in the discussion, please comment on the utility of rinsing fluid in prognostic decision-making for clinicians. Additionally, rinsing fluid monitoring may also be a tool for improving NPWT technology. For instance, it has been reported that the material used commonly for NPWTi (Veraflo) sequestered more inflammatory cytokines and reduced inflammation in macrophages compared to conventional NPWT dressing material (Granufoam) (Veerasubramanian et al., PMCID: PMC8773312).
Minor remarks:
1. Minor text editing issues – Incorrect spacing in line 57, and a typographical error in line 58 that misspells the month “May” are to be corrected. In line 64, the name of the NPWTi setup should be corrected to read as V.A.C. VERAFLO™ (trademark symbol to be used appropriately). In line 67, recommend changing “frozen at” -80 °C, instead of “frozen by”.
2. In table 1, standard notations for bacteria nomenclature should be used. Additionally, the scientific name should be italicized. For example, either S. aureus or Staphylococcus aureus should be used instead of St. aureus.
3. In lines 91-92, please clarify if in this sentence “…median increase to 1543 pg/ml on day two and 861 pg/ml on day four..”, did you mean days one and three, instead of two and four?
4. Line 187 – typo in “polymorphonuclear cells”
5. Figures – need consistent labeling of the y-axis cytokine labels in figures 3, 4, 11, 13, 14 and 15. Use “b-FGF”, “VEGF” and “TNF-a” as these have been used throughout the text.
6. Figures – what do the stars and circles represent (outliers/confidence interval)? Also, indicate the test used for calculating significance of data in the figure legend. Consider replacing “Star (*) is indicating significance.” with “Asterisk (*) indicates significance.”
7. In the materials and methods section, please include the names of the statistical analysis performed.
Author Response
Dear Editor, dear Reviewer,
Again, we would like to thank the reviewers for the helpful comments. We meticulously worked through the review and hope to meet the demands of the reviewers. We added the proposed comments and a few references to more explain our introduction, methodology and conclusion.
In the following paragraphs, we would like to take the opportunity to address all comments by the reviewers in a point-by-point manner (the changes we made to the manuscript are highlighted in red letters in the revised version):
Point-by-point reply to the reviewer’s comments:
Reviewer 2
Major remarks:
- The introduction is bare-bones, and needs to be expanded more to help orient the reader better on the topic. Suggested items to include are:
- Why were the cytokines that were studied in this manuscript chosen (roles in inflammation, repair, etc.)?
Author reply: Thank you for this suggestion. This is true and we added a section explaining the choice of our cytokines. “As until now the cytokine content within the rinsing fluid has not been determined, we chose IL-6, IL-8, b-FGF, TNF-a, VEGF as these depict the central agents in regard to chemotaxis, inflammation and fibroblast recruitment within cutaneous wound healing.“
- How does the surgeon/physician benefit from monitoring the rinsing fluid cytokine levels? Author reply: Thank you for this comment. This is probably one oft he most important questions and we added this to our introduction. „The physician then might be able to react accordingly, for example change a dressing if rising cytokines indicate an infection“
- Include and comment on other studies such as Faes et al. (PMCID: PMC7790174) that has measured inflammatory cytokine levels or cellular composition during conventional negative pressure therapy, also highlighting how this study differs from them.
Author reply: Thank you very much for this interesting and well outlined study. We included this in our manuscript and commented accordingly. „Conventional negative pressure therapy in open abdominal wounds showed distinct clearance patterns of cytokines when compared with serum levels allowing to investigate abdominal sepsis“
- How do co-morbidities and microbial infection play into NPWT therapy and wound healing?
Thank you, this is a central question of high importance. We added the according literature.
„Literature suggests the instillation as a key factor for superior results reducing the bioburden and accelerate wound healing“
- It is suggested that the number of figures be brought down from 15, by clubbing several figures together as panels of one figure. For instance, Figs. 1-5 can be made into 5 panels of Fig. 1.
Author reply: Thank you for this important notation, as this truely gives a better overview over the figures. We made a panel of figure 1-5. Thank you.
- In the discussion, can you hypothesize/comment on why rinsate from non-germ wounds show higher Day 0 IL-8, IL-6 and TNF-a, compared to bacterial infected wounds? This is against the expectation that infected wounds would attract greater immune response and inflammation.
Author reply: Thank you for the comment on the results of our manuscript. The infammatory markers seem to rise within the fluid but no significant difference was found on Day 0 within the group as indicated by the missing bar and asterisk *. Therefore, we did not include this in our discussion as a seperate point and hope to meet your expectations.
- This study establishes the usefulness of rinsing fluid as a marker for healing progression. Therefore, in the discussion, please comment on the utility of rinsing fluid in prognostic decision-making for clinicians. Additionally, rinsing fluid monitoring may also be a tool for improving NPWT technology. For instance, it has been reported that the material used commonly for NPWTi (Veraflo) sequestered more inflammatory cytokines and reduced inflammation in macrophages compared to conventional NPWT dressing material (Granufoam) (Veerasubramanian et al., PMCID: PMC8773312). Author reply: Thank you for this comment on the discussion/ conclusion. We added our section accordingly.
„Physicians should be able to determine the timing of the dressing change and detect superinfections by analyzing the rinsing fluid. Similar technologies like conventional NPWT may also be further investigated as wound fluid is also available“
Minor remarks:
- Minor text editing issues – Incorrect spacing in line 57, and a typographical error in line 58 that misspells the month “May” are to be corrected. In line 64, the name of the NPWTi setup should be corrected to read as V.A.C. VERAFLO™(trademark symbol to be used appropriately). In line 67, recommend changing “frozen at” -80 °C, instead of “frozen by”.
Author reply: Thank you for the comment. We corrected all mistakes.
- In table 1, standard notations for bacteria nomenclature should be used. Additionally, the scientific name should be italicized. For example, either aureusor Staphylococcus aureus should be used instead of St. aureus.
Author reply: Thank you, all corrected, however the journal may change italics.
- In lines 91-92, please clarify if in this sentence “…median increase to 1543 pg/ml on day two and 861 pg/ml on day four..”, did you mean days one and three, instead of two and four?
Author reply: IL-6 showed an undulant course with a significant median increase to 1543 pg/ml on day two and 861 pg/ml on day three (p=0,01 and p=0,04, respectively) compared to the first measurement.
- Line 187 – typo in “polymorphonuclear cells”, Author reply: Already corrected by journal.
- Figures – need consistent labeling of the y-axis cytokine labels in figures 3, 4, 11, 13, 14 and 15. Use “b-FGF”, “VEGF” and “TNF-a” as these have been used throughout the text.
- Figures – what do the stars and circles represent (outliers/confidence interval)? Also, indicate the test used for calculating significance of data in the figure legend. Consider replacing “Star (*) is indicating significance.” with “Asterisk (*) indicates significance.”
Author reply:
Thank you very much. We changed this accordingly.
- In the materials and methods section, please include the names of the statistical analysis performed
Author reply:
Statistical analysis was performed using the Wilcoxon Test for nonparametric paired values over several days, the Mann-Whitney-U-Test for nonparametric values on the same day and the Kruskal-Wallis-Test for comparison of more than two medians. A p-value <0.05 indicated significance and the “Statistical Package for the Social Sciences” (SPSS Inc., Chicago, IL, USA) Version 25.0 was utilized for calculation.